# Evaluation of the Antibacterial Material Production in the Fermentation of *Bacillus*
*amyloliquefaciens-9* from Whitespotted Bamboo Shark (*Chiloscyllium plagiosum*)

**DOI:** 10.3390/md18020119

**Published:** 2020-02-18

**Authors:** Wenjie Zhang, Ling Wei, Rong Xu, Guodong Lin, Huijie Xin, Zhengbing Lv, Hong Qian, Hengbo Shi

**Affiliations:** 1Zhejiang provincial key Laboratory of Silkworm Bioreactor and Biomedicine, College of Life Sciences and medicine, Zhejiang Sci-Tech University, Hangzhou 310018, China; 2College of Animal Science, Zhejiang University, Hangzhou 310015, China

**Keywords:** antibacterial materials, *Bacillus amyloliquefaciens*, whitespotted bamboo shark, biological additive

## Abstract

*Bacillus amyloliquefaciens-9* (GBacillus-9), which is isolated from the intestinal tract of the white-spotted bamboo shark (*Chiloscyllium plagiosum*), can secrete potential antibacterial materials, such as β-1,3-1,4-glucanase and some antimicrobial peptides. However, the low fermentation production has hindered the development of GBacillus-9 as biological additives. In this study, the Plackett–Burman design and response surface methodology were used to optimize the fermentation conditions in a shake flask to obtain a higher yield and antibacterial activity of GBacillus-9. On the basis of the data from medium screening, M9 medium was selected as the basic medium for fermentation. The data from the single-factor experiment showed that sucrose had the highest antibacterial activity among the 10 carbon sources. The Plackett–Burman design identified sucrose, NH_4_Cl, and MgSO_4_ as the major variables altering antibacterial activity. The optimal concentrations of these compounds to enhance antibacterial activity were assessed using the central composite design. Data showed that sucrose, NH_4_Cl, and MgSO_4_ had the highest antibacterial activities at concentrations of 64.8, 1.84, and 0.08 g L^−1^, respectively. The data also showed that the optimal fermentation conditions for the antibacterial material production of GBacillus-9 were as follows: Inoculum volume of 5%, initial pH of 7.0, temperature of 36 °C, rotating speed of 180 rpm, and fermentation time of 10 h. The optimal fermentation medium and conditions achieved to improve the yield of antibacterial materials for GBacillus-9 can enhance the process of developing biological additives derived from GBacillus-9.

## 1. Introduction

The wide usage of antibiotics in the world has rapidly increased the risk of antibiotic-resistant pathogens. Many new and nonconventional anti-infective therapies have been developed and/or identified to solve this threat [1]. Marine microorganisms are regarded as major sources of antimicrobial agents [2]. *Bacillus amyloliquefaciens*, a widely distributed and environmentally friendly aerobic Gram-positive *Bacillus* bacterium, is rich in metabolites and can secrete bacteriostatic substances, such as lipopeptides [3,4] and glycosidases [5]. *B. amyloliquefaciens* has been used in animal feeding [6] or the therapy of plant diseases due to its ability to produce broad-spectrum antibacterial activity [7,8]. The *B. amyloliquefaciens* BTSS-3 isolated from a deep-sea shark (*Centroscyllium fabricii*) has shown antimicrobial activity against pathogenic bacteria, including *Salmonella typhimurium*, *Proteus vulgaris*, *Clostridium perfringens*, and *Staphylococcus aureus* [9]. Isolated from soil, *B. amyloliquefaciens* MET0908 is used to cure cucumber anthracnose [7]. *B. amyloliquefaciens* PPCB004 has inhibitory effects against the fungal pathogens of harvested citrus [8]. *B. amyloliquefaciens* US 573 is reported as a feeding additive that promotes animal nutrient absorption and feed conversion ratio [6].

The whitespotted bamboo shark (*Chiloscyllium plagiosum*) is a demersal cartilage fish distributed from the Indian Ocean to the Western Pacific Ocean. The bacteria in the shark, particularly those in the digestive tract, produce inhibitory compounds responsible for controlling the colonization of potential pathogens in fish [10,11]. *B. amyloliquefaciens-9* (GBacillus-9, also named *Bacillus* sp. GFP-2), a strain of *B. amyloliquefaciens*, was isolated from the intestinal tract of the whitespotted bamboo shark (*C. plagiosum*) [12]. The antimicrobial peptides (AMPs) and β-1,3-1,4-glucanase expressed in GBacillus-9 could partially contribute to inhibit Gram-negative/positive bacteria [12]. In vivo, GBacillus-9 improves immunity by promoting the lysozyme content of the skin mucosa in hybrid sturgeon (*Acipenser sinensis*), indicating GBacillus-9’s potential as a feed additive for aquaculture farmed fish [13]. Furthermore, the data on Japanese eel support the idea of developing bacterial additives in fish raising by using GBacillus-9 [14]. All these data suggest that the antibacterial material in GBacillus-9 can be used as a “drug” in the fish farm, which can help decrease the use of antibiotics in agriculture.

However, the idea of developing a “drug” by using GBacillus-9 is hindered by its low antibacterial material production. The optimization of fermentation conditions will benefit the use of GBacillus-9. The single-factor experiment, Plackett–Burman (PB), and central composite designs (CCDs) [15,16,17,18,19] were used to optimize the fermentation conditions of GBacillus-9, achieve high antibacterial activity, and overcome the limited fermentation efficiency of GBacillus-9.

## 2. Results

### 2.1. Effect of Different Culture Media and Main Ingredient Sources on Antibacterial Activity

Seven basic media with different carbon and nitrogen sources were assessed for the GBacillus-9 culture to determine the culture medium with high efficiency. As shown in Figure 1a, GBacillus-9 had the highest antibacterial activities against *Escherichia coli* TG1 (TG1) and *Bacillus subtilis* BS168 (BS168) when cultured in M9 medium, followed by nutrient yeast beef glucose (NYBD), beef peptone yeast (BPY), nutrient agar (NA), and Luria-Bertani (LB) media. The use of yeast sucrose peptone (YSP) and yeast peptone glucose (YPG) media resulted in GBacillus-9’s lowest antibacterial activity against TG1 and BS168. Thus, M9 medium was selected as the basic culture medium. The growth curve showed that a high GBacillus-9 density did not result in high antibacterial activity. No significant relationship between bacteriostatic activity and growth was observed. 

Carbon and nitrogen sources have significant effects on cell metabolite activity [20]. In the current study, nine carbon sources in M9 medium were screened to culture GBacillus-9 (Figure 1b,c). The maximum antibacterial activity was detected when sucrose was used as the carbon source, followed by glucose, mannitol, and dextrin. The use of maltose, lactose, and soluble starch resulted in small inhibition zones. The addition of citric acid and xylose almost destroyed the antibacterial ability of GBacillus-9.

Seven nitrogen sources were assessed based on the antibacterial activity. As shown in Figure 1c, NH_4_Cl, yeast extract, and peptone had similar antibacterial effects against TG1 and BS168. The addition of urea and beef cream led to the loss of antibacterial activity. NH_4_Cl was selected as the nitrogen source because of its advantages, such as low cost and source availability.

### 2.2. Determination of the Initial Concentration of Each Component in M9 Medium Using Single-Factor Experiment Design

The component in M9 medium was altered to determine the antibacterial activity of the GBacillus-9 strain in the modified M9 medium. As shown in Figure 2, increasing each component increased the antibacterial activity of GBacillus-9. The optimum concentrations of sucrose, Na_2_HPO_4_, KH_2_PO_4_, NH_4_Cl, MgSO_4_, and NaCl were at 30, 8.5, 3.0, 1.0, 0.5, and 0.5 g L^−1^, respectively.

### 2.3. Sucrose, NH_4_Cl, and MgSO_4_ as Key Factors Affecting GBacillus-9 Culture 

The results of various cultures with antibacterial activity are shown in Table 6. After analyzing the samples from culture media, the mean diameters of the inhibition zones in all samples were measured. All data were used as dependent variables Y1 (mean inhibition zone diameter of TG1) and Y2 (mean inhibition zone diameter of BS168) in the PB design.

The polynomial model, which predicted the inhibition zone diameter of TG1 obtained from PB regression analysis, was expressed in terms of coded factors as
Y1(cm) = 1.362 + 0.121*A + 0.030*B + 0.045*C + 0.1203*D + 0.004*E − 0.072*F.(1)

The second response value analyzed was the inhibition zone diameter of BS168. The polynomial model, which predicted the inhibition zone diameter of BS168 obtained from PB regression analysis, was expressed in terms of coded factors as
Y2(cm) = 1.416 + 0.184*A + 0.134*B+0.016*C + 0.124*D + 0.058*E − 0.070*F.(2)

The analysis of variance (ANOVA) was applied to test the availability of PB design for the responses Y1 and Y2. Results are shown in Table 1 and Table 2. The analysis of *P*-values and the Pareto chart (Figure 3) showed that among the variables tested, sucrose, NH_4_Cl, and MgSO_4_ had significant (*p* < 0.05) effects on the inhibition zone diameter of TG1 (Table 1, Figure 3a), whereas only sucrose had a significant (*p* < 0.05) effect on the inhibition zone diameter of BS168 (Table 2, Figure 3b).

### 2.4. Optimization of Fermentation Conditions by Using CCD

CCD was utilized to assess the influence of three individual factors, namely, sucrose (A), NH_4_Cl (B), and MgSO_4_ (C), along with their interaction effects on the yield of antibacterial materials. The experimental and predicted values of the inhibition zone diameter under different treatment conditions are shown in Table 8. The predicted responses Y3 (TG1) and Y4 (BS168) for inhibition zone diameter in terms of coded factors are expressed as follows:Y3 (cm) = 1.4307 + 0.11103*A + 0.06766*B − 0.06860*C + 0.02349A^2^ − 0.02070*B^2^ − 0.02070*C^2^ + 0.0362*AB − 0.0263*AC − 0.0338*BC.(3)
Y4 (cm) = 1.44359 + 0.09386*A + 0.05295*B − 0.05951*C + 0.00069A^2^ − 0.02582B^2^ − 0.02406C^2^ + 0.02375*AB − 0.00625*AC − 0.02875*BC.(4)
where Y3 and Y4 are the response variables of inhibition zone diameter, and A, B, and C are independent variables in coded units. The regression model was statistically detected using the F-test, and ANOVA was utilized to evaluate the significance and adequacy of the model [21].

As shown in Table 3 and Table 4, the *p*-values obtained using ANOVA and the F-test were less than 0.05, demonstrating that the model term was significant (*p* < 0.05). Table 3 and Table 4 showed that the coefficients for A, B, and C were extremely significant (*p* < 0.05), indicating that sucrose, NH_4_Cl, and MgSO_4_ had a remarkable influence on the antibacterial production of GBacillus-9. The model was significant with a very low *p*-value < 0.01 (*p*-value Probability > F). The R^2^ and adjusted R^2^ values for TG1 (0.9748 and 0.9522, respectively) and BS168 (0.9773 and 0.9569, respectively) were close to 1, indicating the high efficacy of Equations (3) and (4) [22].

The three-dimensional response surfaces and contour plots illustrated in Figure 4 depict the interactions of sucrose, NH_4_Cl, and MgSO_4_ on antibacterial material yield by using TG1 and BS168 as substrates [23,24]. These results also directly show the response over a region of independent variables and the relationship between the experimental levels of each factor. Figure 4a reveals the effects of sucrose and NH_4_Cl on the antibacterial material yield of GBacillus-9, indicating that the inhibition zone increased upon increasing sucrose and NH_4_Cl. The maximum inhibition zone diameter was observed when the concentrations of sucrose and NH_4_Cl were at 64.8 and 1.84 g L^−1^, respectively. Subsequently, the inhibition zone diameter of TG1 declined with the increase in sucrose and NH_4_Cl concentrations. Increasing the concentration of MgSO_4_ to more than 0.08 g L^−1^ resulted in a decreased inhibition zone diameter. The effect of MgSO_4_ on inhibition zone diameter is displayed in Figure 4c, which had a trend similar to those in Figure 4a,b. The inhibition zone diameter of BS168 was smaller than that of TG1 under the same culture medium, as shown in Figure 4d–f, but a similar antibacterial effect trend was observed in Figure 4a–c. Finally, the optimal concentrations of sucrose, NH_4_Cl, and MgSO_4_ in the culture medium for GBacillus-9 were 64.8, 1.84, and 0.08 g L^−1^, respectively. Under the optimum conditions, the predicted maximum diameters of TG1 and BS168 inhibition zones were 1.93 and 1.82 cm, respectively.

### 2.5. Optimization of Fermentation Parameters

The fermentation time, temperature, inoculum volume, initial pH, and rotation speed were also assessed to further explore the optimal fermentation conditions of GBacillus-9. As shown in Figure 5, the inhibition zone diameter of anti-TG1 and anti-BS168 showed a similar pattern, that is, the inhibition zone diameter increased with fermentation time, temperature, inoculum volume, initial pH, and rotation speed. The maximum inhibition zone diameters of anti-TG1 and anti-BS168 were achieved when the fermentation time, temperature, inoculum volume, initial pH, and rotation speed were 10 h, 36 °C, 5%, 7.0, and 180 rpm, respectively. Subsequently, the inhibition zone diameter of TG1 declined with fermentation time, temperature, inoculum volume, initial pH, and rotation speed.

### 2.6. Fermentation Using a 20 L Bioreactor

The optimized fermentation conditions (inoculum volume of 5%, initial pH of 7.0, rotating speed of 180 rpm, and temperature of 36 °C) was verified using a 20 L bioreactor to determine whether the results predicted by the model were consistent with the actual results. The fermentation product was subjected to an antibacterial activity test. Figure 6 showed that the culture supernatant of GBacillus-9 had inhibition zone diameters of 1.82 and 1.78 cm for TG1 and BS168, respectively. These results were in agreement with the predicted values.

## 3. Discussion

The marine environment is a largely untapped source for the isolation of new microorganisms with the potential to produce bioactive compounds [25]. The host organism synthesizes bioactive compounds as primary or secondary metabolites to protect or maintain homeostasis [26]. GBacillus-9, which is isolated from the intestine of the whitespotted bamboo shark, has shown potential in the development of biological additives [12,13]. Thus, investigating the optimal fermentation conditions with high efficiency and low cost is an important strategy to develop biological additives. In the current study, PB design and CCD were applied to evaluate the fermentation efficiency of GBacillus-9 through various media, and the results showed efficient fermentation parameters with the improved yield of the antibacterial material in GBacillus-9.

The GBacillus-9 strain has shown a broad-spectrum antibacterial activity on Gram-negative and Gram-positive bacteria [12]. In the current study, TG1 and BS168, which represented the Gram-negative and the Gram-positive bacteria, respectively, were used to detect antibacterial activity. The current data showed that the antibacterial material secreted by GBacillus-9 had a stronger activity against Gram-negative bacteria (TG1) than Gram-positive bacteria (BS168), which agreed with the fact that AMPs blocked the growth of Gram-negative bacteria by entering the membrane and disrupting the local electrostatic fields [27]. The weak activity of GBacillus-9 against Gram-positive bacteria may be due to the inherent properties of Gram-positive bacteria, which contain more peptidoglycan than Gram-negative bacteria. The peptidoglycan hinders the entry of AMPs into cells, thereby reducing antibacterial activity [28,29]. 

The medium most beneficial to the production of antibacterial substances in GBacillus-9 was obtained by screening different media. No evident relationship was found between the bacteriostatic activity and the number of bacteria. The increased antibacterial activity may be caused by a secondary metabolite secreted by GBacillus-9. 

There have been many reports on the antibacterial substances of *B. subtilis,* which mainly achieved the antibacterial effect by secreting peptides, lipids, and other substances [30,31]. Most of the *B. subtilis* strains capable of secreting antibacterial substances are isolated from soil and plants [6,7,8]. However, it is still relatively rare to isolate *Bacillus subtilis* in the ocean, and the GBacillus-9 culture medium isolated in the present study has a strong antibacterial activity against Gram-positive and Gram-negative bacteria. Moreover, after optimizing the culture medium, the high activity remained even after spray drying, which was conducive to large-scale market applications. 

Carbon and nitrogen sources play an important role in the growth and cellular metabolism of bacteria [32,33]. Although these carbon sources resulted in the similar OD value of GBacillus-9, sucrose, glucose, and mannitol had the highest antibacterial activity. The various antibacterial material products with different carbon sources agreed with the suggestion that high cell weight does not always lead to the increased production of antibacterial substances [34,35]. Sucrose was selected because it was cheaper than glucose and mannitol. In the present study, the effects of nine carbon sources and seven nitrogen sources on the secretion of GBacillus-9 antibacterial substances were screened. Unsurprisingly, sucrose and NH_4_Cl were identified as the most important factors with respect to inhibition zone diameters of anti-TG1 and anti-BS168. Both sources met the bacterial mass and the production of antibacterial substances. 

The PB design is a screening procedure that identifies essential variables in the production of a response parameter by analyzing the main effects, which allow multiple factors to be screened within one experiment. This design has been well used in assessing the nutritional requirements of bacteria to optimize enzyme production [36,37,38]. In the present study, sucrose, NH_4_Cl, and MgSO_4_ were identified as three key factors affecting the yield of anti-TG1. However, only sucrose was identified as the key factor affecting the yield of anti-BS168. This difference was attributed to the intrinsic characteristics, e.g., the composition of cytoderm, and of Gram-positive and Gram-negative bacteria. 

The increase in sucrose concentration increased the antibacterial activity of GBacillus-9, suggesting an increased production of bacteria when the concentration of sucrose was higher than 46.8 g L^−1^. The reason for using sucrose at 46.8 g L^−1^ in the culture medium was to avoid the viscosity caused by the higher concentration of sucrose in the subsequent spray drying for bacterial preparation [39]. Furthermore, a high concentration of MgSO_4_ inhibited the growth of GBacillus-9.

Aside from the cultured medium, the fermentation process also greatly affected the production. Moreover, pH can affect the cellular enzyme activity and the stability of microbial cell membranes, altering membrane permeability and nutrient absorption [40,41,42]. Additionally, inoculation affects the growth of microorganisms and the accumulation of metabolites. Excessive inoculation accelerates the consumption of the medium, whereas low-inoculum cell concentrations prolong the log phase [43,44], both of which reduce the efficiency of fermentation. Besides, culture temperature, rotation speed, and culture time affect the secretion after fermentation in different degrees [45]. In the present study, the optimal inoculum volume, temperature, rotation speed, and initial pH for GBacillus-9 were 5% (v/v), 36 °C, 180 rpm, and 7.0, respectively, which were consistent with those of other *B. amyloliquefaciens* strains [46,47]. 

The limitation of the present study was that the antibacterial materials secreted by GBacillus-9 were not completely addressed. However, our previous data showed that β-1,3-1,4-glucanase and the antimicrobial peptides secreted by GBacillus-9 contributed to its antimicrobial activity [12]. The detection of β-1,3-1,4-glucanase in the crude proteins from GBacillus-9 culture supernatant confirmed that GBacillus-9 could indeed express β-1,3-1,4-glucanase. Furthermore, the antimicrobial activity of purified recombinant β-1,3-1,4-glucanase cloned from the GBacillus-9 genome suggests that β-1,3-1,4-glucanase is a factor contributing to antimicrobial activity [12]. The genome data indicated that GBacillus-9 contained genes encoding LCI (a 47 residue cationic antimicrobial peptide), yellowfin tuna GAPDH-related antimicrobial peptide, and human GAPDH-related AMPs [12]. Beyond the potential AMPs, metabolites, such as difficidin, bacillibactin, bacilysin, surfactin, butirosin, macrolactin, bacillaene, and fengycin, and bacteriocins have been predicted in the genome of GBacillus-9 [12], which may contribute to the antibacterial role. Collectively, these results suggest that the inhibitory effects of GBacillus-9 on Gram-negative/positive bacteria might be dependent on complex metabolites. More experiments are needed to address the antibacterial materials in GBacillus-9 in the future. 

The inhibition zone diameter of the anti-TG1 and anti-BS168 in the optimal culture medium reached 1.82 and 1.78 cm, respectively, which were significantly improved compared to that in the LB medium. The finding that the bacteriostatic test of GBacillus-9 with 20 L was consistent with the predicted regression model suggested success in using the response surface method to improve the fermentation efficiency and that the factor effects were real. 

In the present study, optimization studies using PB and response surface methodology identified the optimal nutritional factors involved in the maximal production of antibacterial compounds from GBacillus-9, supporting a fermenter to obtain a large number of GBacillus-9. Thus, two necessary market application requirements for the easily obtained and low-cost development of a “drug” were met. These data can enhance the process of developing biological additives. In addition, results showed the optimal fermentation time, temperature, inoculum volume, initial pH, and rotation speed for GBacillus-9. However, the antibacterial material of GBacillus-9 and specific mechanism behind its biological activity remain unknown. Hence, further research is required.

## 4. Materials and Methods 

### 4.1. Strain and Media

*B. amyloliquefaciens-9* (GBacillus-9; CGMCC number: 13337, accession number: CP021011) [12] was isolated from the intestinal tract of *C. plagiosum*, which can secrete a variety of digestive enzymes to inhibit the growth of pathogenic bacteria. Seven different synthetic media were screened to improve the antimicrobial substance production of GBacillus-9. The media included LB (10 g L^−1^ peptone, 10 g L^−1^ NaCl, and 5 g L^−1^ yeast extract), BPY (5 gL^−1^ glucose, 1 g L^−1^ peptone, 5 g L^−1^ beef extract, 5 g L^−1^ NaCl, and 5 g L^−1^ yeast extract), NYBD (8 g L^−1^ beef extract, 5 g L^−1^ yeast extract, and 10 g L^−1^ glucose), NA (10 g L^−1^ peptone, 2.5 g L^−1^ glucose, and 3 g L^−1^ beef extract), YPG (10 g L^−1^ yeast extract, 20 g L^−1^ peptone, and 20 g L^−1^ glucose), YSP (5 g L^−1^ yeast extract, 20 g L^−1^ sucrose, and 10 g L^−1^ peptone), and M9 (20 g L^−1^ glucose, 8.5 g L^−1^ Na_2_HPO_4_, 3 g L^−1^ KH_2_PO_4_, 1 g L^−1^ NH_4_Cl, 0.5 g L^−1^ NaCl, and 0.5 g L^−1^ MgSO_4_) media. The medium pH was adjusted to 7.0–7.2. GBacillus-9 was cultured for 12 h at 37 °C with rotary shaking at 220 rpm in different culture media, and the antimicrobial ability was detected. All the experiments were performed at least thrice.

### 4.2. Detection of Antimicrobial Activity

The antimicrobial ability of GBacillus-9 was identified using the inhibition zone method [12]. The Gram-positive bacterium BS168 and the Gram-negative bacterium TG1 were used as test strains and inoculated into LB solid medium at 1% of the total amount. After solidification, an Oxford cup was placed on the medium and added with a volume of 250 μL fermentation product. The sample was diluted 1000 times by using 50 mg/mL ampicillin (Amp) as the positive control and the blank medium as the negative control. After incubation for 10 h in a 37 °C incubator, the diameter of the inhibition zone (accurate to 0.1 mm) was measured.

### 4.3. Effect of Carbon and Nitrogen Sources

GBacillus-9 was grown in M9 basal liquid medium. The basal liquid medium was supplemented separately with carbon sources (20 g L^−1^), namely, glucose, xylose, sucrose, dextrin, citric acid, mannitol, maltose, lactose, and soluble starch. In the case of nitrogen (1 g L^−1^), various nitrogen sources, namely, peptone, urea, NH_4_Cl, yeast extract, (NH_4_)_2_SO_4_, yeast extract cream, and beef cream, were supplemented. GBacillus-9 was cultured at 37 °C with rotary shaking at 220 rpm in different culture media and incubated for 10 h at 37 °C. The antimicrobial ability was detected. All the experiments were performed at least thrice.

### 4.4. Single-Factor Experiment Design

The experiments were carried out by varying one condition at a time [48]. First, the carbon source (sucrose) was set to five concentration gradients (10–50 g L^−1^), and GBacillus-9 was cultured in M9 culture media for 10 h at 37 °C. Next, the antimicrobial ability was detected using the inhibition zone method to find the optimal concentration of the carbon source. For NH_4_Cl, the same M9 basal liquid medium was employed, but the NH_4_Cl was adjusted to a concentration of 0.1–2.0 g L^−1^. Other factors, such as Na_2_HPO_4_, KH_2_PO_4_, NaCl, and MgSO_4_, were also set to concentration gradients of 6.5–10.5, 1.0–5, 0.1–2.0, and 0.1–2.0 g L^−1^, respectively. The antimicrobial activity of supernatant was then detected to find the optimal concentration. All the experiments were done in triplicate.

### 4.5. Screening of Significant Variables Using PB Design 

The relative significance of six variables, namely, sucrose, Na_2_HPO_4_, KH_2_PO_4_, NH_4_Cl, MgSO_4_, and NaCl, were investigated using the PB design (Table 5 and Table 6) to determine the key factors significantly affecting the antibacterial activity of GBacillus-9. Six variables of medium composition were tested at low (−1) and high (+1) levels on the basis of the PB matrix design [49]. The PB experimental design was based on the first-order model (5):Y = β_0_ + Σβ_i_x_i_,(5)
where Y is the predicted response, β_0_ is the model intercept, β_i_ is the linear coefficient, and x_i_ is the level of the independent variable. According to the *P*-value of the experimental results, the main factors affecting the diameter of the inhibition zone can be obtained. The factors with significant (*p* < 0.05) influence on the experimental results were selected for the response surface optimization.

### 4.6. Optimization of Significant Variables Using CCD 

The CCD experiment can achieve the sequentiality of the test and better fits the corresponding surface because, in the CCD test design process, many points are beyond the original level [15,16,50]. A CCD with five coded levels was used for locating the optimum conditions of sucrose, NH_4_Cl, and MgSO_4_ to find the optimal cultivation conditions for GBacillus-9. For the three factors, this trial was essentially a full 23 factorial design with six axial points (a = 1.68) and six replications of the center points, resulting in a total number of 20 experiments (Table 7 and Table 8). Mathematical models describing the relationship among the process variables in terms of their linear, quadratic, and interactive effects used were described using a second-order polynomial equation presented in Equation (6):Y = β_0_ + Σβ_i_x_i_ + Σβ_ii_x_i_^2^ + Σβ_ij_x_i_x_j_,(6)
where Y is the predicted response, β_0_ is the intercept term, β_i_ represents the linear coefficients, β_ii_ represents the quadratic coefficients, β_ij_ represents the interactive coefficients, and x_i_ and x_j_ are the coded independent variables [19,51]. Other media concentrations remained at 8.5_,_ 3, and 0.5 g L^−1^ for Na_2_HPO_4_, KH_2_PO_4_ and NaCl, respectively.

### 4.7. Optimization of Fermentation Conditions

On the basis of the optimized medium using the CCD experiment, a shake flask was used to optimize the fermentation conditions (pH, rotation speed, temperature, inoculum size, and time). For pH, M9 medium was dispensed into 250 mL flasks (100 mL per flask). The flasks were autoclaved at 121 °C for 20 min. After cooling, each flask was inoculated with GBacillus-9 and incubated at different pH values (6–8) for 12 h. All the experiments were done in triplicate. For the other parameters, the same M9 medium was employed, but the medium was adjusted to different values. A total of 100 mL in each treatment was dispensed into the 250 mL flask and replicated thrice.

### 4.8. Preparation and Property Analysis 

A small-scale fermentation experiment by using a 20 L fermenter was carried out to verify the optimal medium formula and fermentation conditions. The antibacterial activity of the fermentation broth was determined, which laid a foundation for preparing the bacterial preparation.

## Figures and Tables

**Figure 1 marinedrugs-18-00119-f001:**
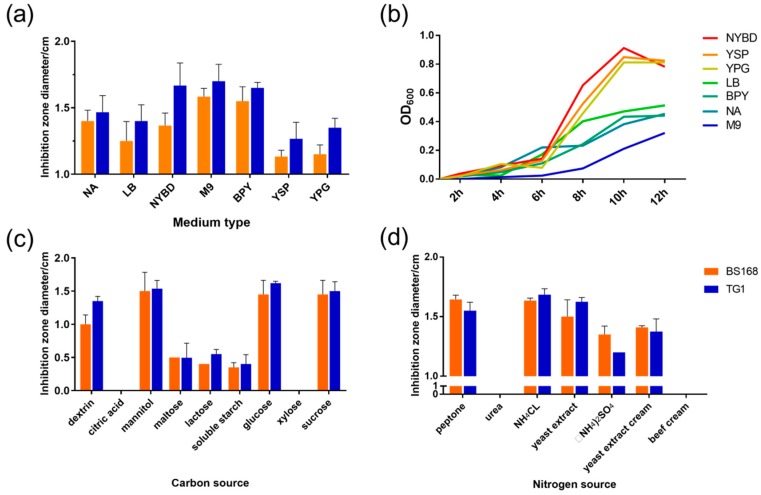
Effects of various media and main ingredient sources on antibacterial activity. (**a**) Effect of different media (nutrient agar (NA), Luria-Bertani (LB), nutrient yeast beef glucose (NYBD), M9, beef peptone yeast (BPY), yeast sucrose peptone (YSP), and yeast peptone glucose (YPG)) on the antibacterial activity of metabolites. TG1 and BS168 (representing the Gram-negative and Gram-positive bacteria, respectively) were used to detect antibacterial activity. The Y axis represents the diameter of the inhibition zone, indicating the antibacterial activity of metabolites. (**b**) Growth curves of GBacillus-9 in different media. Effects of (**c**) carbon and (**d**) nitrogen sources on the antibacterial activity of metabolites in M9 medium. The sources of carbon included dextrin, citric acid, mannitol, maltose, lactose, soluble starch, glucose, xylose, and sucrose. The sources of nitrogen included peptone, urea, NH_4_Cl, yeast extract, (NH_4_)_2_SO_4_, yeast extract cream, and beef cream.

**Figure 2 marinedrugs-18-00119-f002:**
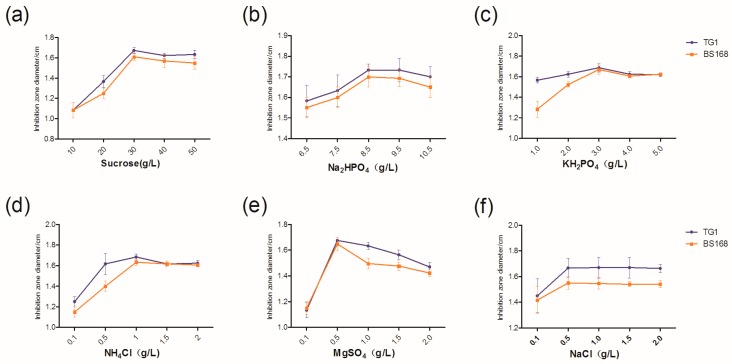
Optimum concentrations of Sucrose (**a**), Na_2_HPO_4_ (**b**), KH_2_PO_4_ (**c**), NH_4_Cl (**d**), MgSO_4_ (**e**), and NaCl (**f**) in M9 medium.

**Figure 3 marinedrugs-18-00119-f003:**
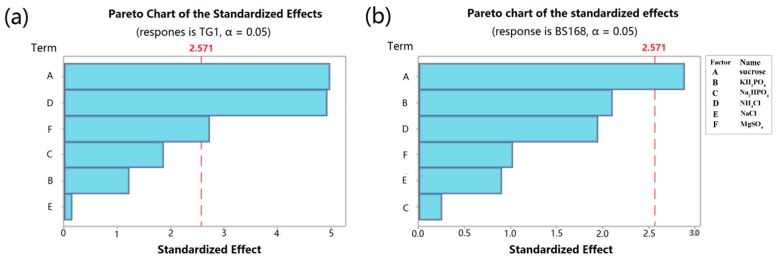
Pareto chart of the standardized effects: (**a**) Response is TG1; (**b**) response is BS 168;

**Figure 4 marinedrugs-18-00119-f004:**
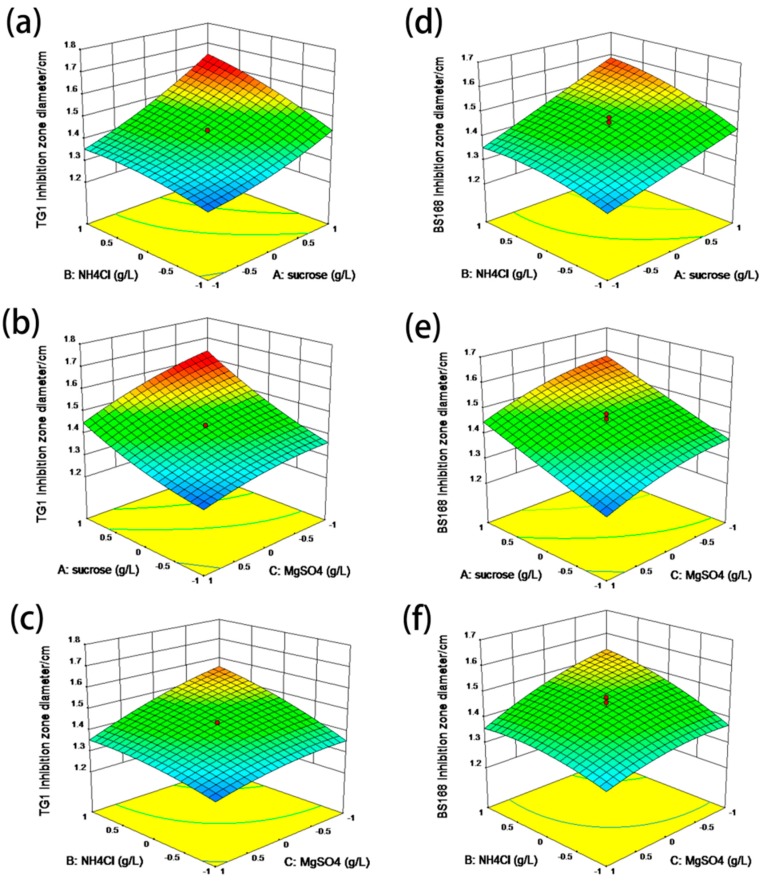
Response surface curves of the inhibition zone diameters of TG1 and BS168 by using the media cultured with GBacillus-9. (**a**) and (**d**): Interaction between sucrose and NH_4_Cl. (**b**) and (**e**): Interaction between sucrose and MgSO_4_. (**c**) and (**f**): Interaction between NH_4_Cl and MgSO_4_. The standard concentrations of Na_2_HPO_4_, KH_2_PO_4_, and NaCl used in the experiment were 8.5, 3, and 0.5 g L^−1^, respectively.

**Figure 5 marinedrugs-18-00119-f005:**
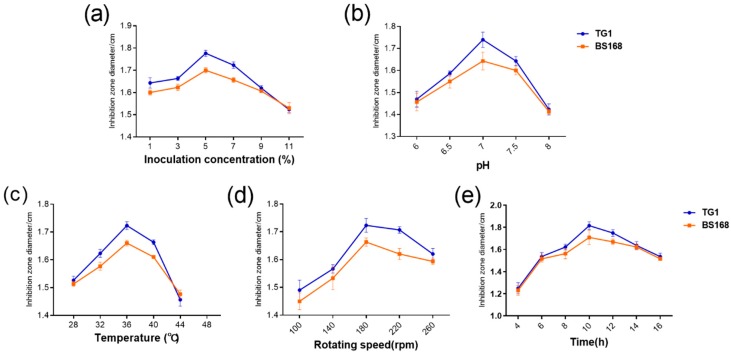
Effect of fermentation parameters ((**a**): Inoculation concentration, (**b**): pH, (**c**): Temperature, (**d**): Rotating speed, and (**e**): Time) on the antibacterial activity of metabolites.

**Figure 6 marinedrugs-18-00119-f006:**
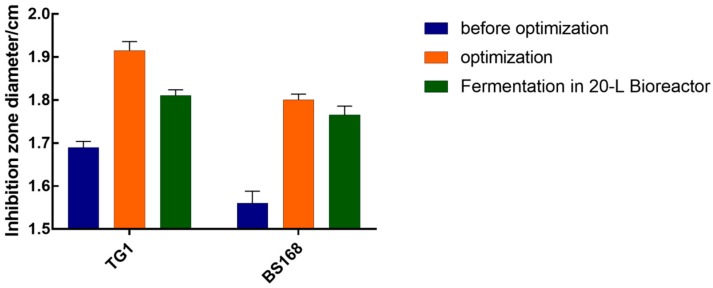
Validation of the optimal fermentation conditions of GBacillus-9 in 20 L bioreactor.

**Table 1 marinedrugs-18-00119-t001:** Analysis of variance (ANOVA) for the regression model of the inhibition zone of TG1 using the Plackett–Burman design.

Variable	Coefficient Estimate	SE Coefficient	T Value	*p* Value
constant	1.362	0.024	55.900	0.000
sucrose	0.121	0.024	4.980	0.004 *
Na_2_HPO_4_	0.030	0.024	1.220	0.277
KH_2_PO_4_	0.045	0.024	1.860	0.122
NH_4_Cl	0.120	0.024	4.940	0.004 *
NaCl	0.004	0.024	0.150	0.888
MgSO_4_	−0.072	0.026	−2.730	0.041 *

* *p* < 0.05, Significance was declared when *p* < 0.05.

**Table 2 marinedrugs-18-00119-t002:** Analysis of variance (ANOVA) for the regression model of the inhibition zone of BS168 using the Plackett–Burman design.

Variable	Coefficient Estimate	SE Coefficient	T Value	*p* Value
constant	1.416	0.063	22.230	0.000
sucrose	0.184	0.063	2.890	0.034 *
Na_2_HPO_4_	0.134	0.063	2.110	0.814
KH_2_PO_4_	0.016	0.063	0.250	0.814
NH_4_Cl	0.124	0.063	1.950	0.109
NaCl	0.058	0.063	0.900	0.408
MgSO_4_	−0.070	0.068	−1.020	0.335

* *p* < 0.05, Significance was declared when *p* < 0.05.

**Table 3 marinedrugs-18-00119-t003:** Analysis of CCD test results for TG1.

Source	Degree of Freedom	Adj SS	Adj MS	Value	*p*-Value
Model	9	0.34	0.038	43.02	<0.0001 ^#^
A	1	0.17	0.17	190.38	<0.0001 ^#^
B	1	0.063	0.063	70.70	<0.0001 ^#^
C	1	0.064	0.064	72.66	<0.0001 ^#^
A2	1	0.011	0.011	11.89	0.0062 ^#^
B2	1	5.512 × 10^−3^	5.512 × 10^−3^	6.23	0.0316 ^#^
C2	1	9.112 × 10^−3^	9.112 × 10^−3^	10.30	0.0093 ^#^
A*B	1	7.953 × 10^−3^	7.953 × 10^−3^	8.99	0.0134 ^#^
A*C	1	6.177 × 10^−3^	6.177 × 10^−3^	6.98	0.0246 ^#^
B*C	1	6.177 × 10^−3^	6.177 × 10^−3^	6.98	0.00246 ^#^
Residual	10	8.844 × 10^−3^	8.844 × 10^−4^		
Lack of fit	5	8.644 × 10^−3^	1.729 × 10^−3^	43.22	<0.0004 ^#^
Pure error	5	2.000 × 10^−4^	4.000 × 10^−5^		
Cor Total	19	0.35			

R^2^ = 97.48%; adj R^2^ = 95.22%, ^#^ Model terms are significant

**Table 4 marinedrugs-18-00119-t004:** Analysis of CCD test results for BS168.

Source	Degree of Freedom	Adj SS	Adj MS	Value	*p*-Value
Model	9	0.24	0.026	47.82	<0.0001 ^#^
A	1	0.12	0.12	220.20	<0.0001 ^#^
B	1	0.038	0.038	70.09	<0.0001 ^#^
C	1	0.048	0.048	88.52	<0.0001 ^#^
A2	1	4.513 × 10^−3^	4.513 × 10^−3^	8.26	0.0166 ^#^
B2	1	3.125 × 10^−4^	3.125 × 10^−4^	0.57	0.4669
C2	1	6.612 × 10^−3^	6.612 × 10^−3^	12.10	0.0059
A*B	1	6.905 × 10^−6^	6.905 × 10^−6^	0.013	0.9127
A*C	1	9.611 × 10^−3^	9.611 × 10^−3^	17.59	0.0018 ^#^
B*C	1	8.340 × 10^−3^	8.340 × 10^−3^	15.26	0.0029 ^#^
Residual	10	5.464 × 10^−3^	5.464 × 10^−4^		
Lack of fit	5	2.930 × 10^−3^	5.861 × 10^−4^	1.16	0.4385
Pure error	5	2.533 × 10^−3^	5.067 × 10^−4^		
Cor Total	19	0.24			

R^2^ = 97.73%; adj R^2^ = 95.69%, ^#^ Model terms are significant (*p* < 0.05).

**Table 5 marinedrugs-18-00119-t005:** Plackett–Burman (PB) factor level design.

Factor		Level (g L^−1^)	
		−1	+1
A	sucrose	20.0	40.0
B	KH_2_PO_4_	2.0	4.0
C	Na_2_HPO_4_	7.5	9.5
D	NH_4_Cl	0.5	1.5
E	NaCl	0.25	0.75
F	MgSO_4_	0.25	0.75
G	blank	0.0	0.0

**Table 6 marinedrugs-18-00119-t006:** PB experimental design and results.

Experiment Group	A	B	C	D	E	F	TG1 Inhibition Zone Diameter (cm)	BS168 Inhibition Zone Diameter (cm)
1	−1	1	−1	−1	−1	1	1.000	1.150
2	1	−1	1	1	−1	1	1.500	1.620
3	1	1	−1	1	1	−1	1.700	1.650
4	−1	−1	1	1	1	−1	1.400	1.000
5	−1	−1	−1	−1	−1	−1	1.100	1.000
6	−1	−1	−1	1	1	1	1.200	1.300
7	−1	−1	−1	−1	1	1	1.150	1.200
8	1	1	−1	1	−1	−1	1.650	1.650
9	0	0	0	0	0	0	1.500	1.200
10	−1	−1	1	−1	−1	−1	1.500	1.300
11	1	1	1	1	−1	1	1.300	1.300
12	1	1	1	−1	1	−1	1.200	1.500
13	1	1	1	−1	1	1	1.400	1.600

**Table 7 marinedrugs-18-00119-t007:** CCD experimental design.

Factors		Levels of Variables (g L^−1^)				
		−1.68	−1	0	+1	1.68
A	sucrose	13.200	20.000	30.000	40.000	46.800
B	NH_4_Cl	0.160	0.500	1.000	1.500	1.840
C	MgSO_4_	0.080	0.250	0.500	0.750	0.920

**Table 8 marinedrugs-18-00119-t008:** Results of CCD experimental design.

Test Number	A	B	C	TG1 Inhibition Zone Diameter (cm)		BS168 Inhibition Zone Diameter (cm)	
				Actual Value	Predicted Value	Actual Value	Predicted Value
1	0.00	0.00	0.00	1.43	1.43	1.42	1.44
2	−1.00	1.00	1.00	1.27	1.26	1.25	1.25
3	0.00	0.00	0.00	1.43	1.43	1.43	1.44
4	1.00	1.00	1.00	1.45	1.5	1.44	1.47
5	−1.68	0.00	0.00	1.3	1.31	1.28	1.29
6	0.00	−1.68	0.00	1.25	1.26	1.27	1.28
7	0.00	0.00	0.00	1.42	1.43	1.44	1.44
8	0.00	0.00	0.00	1.43	1.43	1.46	1.44
9	−1.00	−1.00	−1.00	1.31	1.28	1.32	1.30
10	1.00	−1.00	1.00	1.37	1.36	1.38	1.37
11	−1.00	−1.00	1.00	1.25	1.26	1.24	1.25
12	0.00	0.00	1.68	1.27	1.26	1.29	1.28
13	0.00	0.00	0.00	1.43	1.43	1.48	1.44
14	0.00	0.00	−1.68	1.50	1.49	1.47	1.48
15	0.00	1.68	0.00	1.52	1.49	1.48	1.46
16	1.68	0.00	0.00	1.72	1.68	1.62	1.60
17	1.00	1.00	−1.00	1.75	1.76	1.66	1.66
18	1.00	−1.00	−1.00	1.45	1.48	1.44	1.45
19	−1.00	1.00	−1.00	1.38	1.41	1.4	1.41
20	0.00	0.00	0.00	1.44	1.43	1.43	1.44

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
