# Peer review of "Evaluation of the Antibacterial Material Production in the Fermentation of Bacillus amyloliquefaciens-9 from Whitespotted Bamboo Shark (Chiloscyllium plagiosum)"

_marinedrugs, 2020, doi:10.3390/md18020119_

Round 1

Reviewer 1 Report

The authors did the recomendations suggested. Now I think that the paper is appropriate.

Author Response

Thanks for the reviewer's positive comment.

Reviewer 2 Report

The manuscript is well written and presents interesting results. I assume this is a revised version, and I don't see any points to be enhanced. I would suggest the publication of this paper as it is.

Author Response

(The authors gave the same response as above.)

Reviewer 3 Report

The manuscript is very interesting. 

I make some suggestions on the document attached.

I would like the authors to answer the follows issues:

On the Tables 1 and 2, what's means "Contribution"; Please, on the results insert the Pareto chart of standardized effect of regression-model; Maybe the session Materials and Methods could be before the results and improve the quality of the manuscript; Add some reference research on the Materials and Methods as the follow suggestions: Bai, X.; Wen, S.; Liu, J.; Lin, Y. Response Surface Methodology for Optimization of Copper Leaching from Refractory Flotation Tailings. Minerals, 2018, 8, 165. Yang, J.; Shuai, Z.; Zhou, W.; Ma, S. Griding Optimization of Cassiterire-Polymetallic Sulfide Ore. Minerals, 2019, 9, 134. https://www.mdpi.com/1660-3397/17/3/162/htm https://www.mdpi.com/1660-3397/17/1/40/htm https://www.mdpi.com/1660-3397/16/10/372/htm

Author Response

We think the reviewer’ positive and kindly comments. All the comments in the reviewed file were read carefully and revised according the suggestion. All the changes are in yellow in the revised manuscript.

 Point 1:On the Tables 1 and 2, what's means "Contribution";

Response: Thanks for your comments and apologize for this mistake in the text. We have change ‘Contribution’ to ‘SE Coefficient’, which means coefficient standard error. (line 118, 121)

Point 2:Please, on the results insert the Pareto chart of standardized effect of regression-model;

Response: Thanks for your comment. We added a Pareto chart in the result section (Figure 3) accordingly. (line 113, 123).

Point 3:Maybe the session Materials and Methods could be before the results and improve the quality of the manuscript;

Response: Thanks for your comment. But the form of the manuscript is formatted according to the requirement of the Journal.

Point 4:Add some reference research on the Materials and Methods as the follow suggestions:

Bai, X.; Wen, S.; Liu, J.; Lin, Y. Response Surface Methodology for Optimization of Copper Leaching from Refractory Flotation Tailings. Minerals, 2018, 8, 165.

Yang, J.; Shuai, Z.; Zhou, W.; Ma, S. Griding Optimization of Cassiterire-Polymetallic Sulfide Ore. Minerals, 2019, 9, 134.

https://www.mdpi.com/1660-3397/17/3/162/htm; https://www.mdpi.com/1660-3397/17/1/40/htm;

https://www.mdpi.com/1660-3397/16/10/372/htm

Response: Thanks for your comment. We added the four references on the Materials and Methods according to the comment. (line 59, 304, 236. Reference [18], [19], [48], [50])